# A Mixed-Methods Participatory Intervention Design Process to Develop Intervention Options in Immediate Food and Built Environments to Support Healthy Eating and Active Living among Children and Adolescents in Cameroon and South Africa

**DOI:** 10.3390/ijerph191610263

**Published:** 2022-08-18

**Authors:** Agnes Erzse, Teurai Rwafa-Ponela, Petronell Kruger, Feyisayo A. Wayas, Estelle Victoria Lambert, Clarisse Mapa-Tassou, Edwin Ngwa, Susan Goldstein, Louise Foley, Karen J. Hofman, Stephanie Teguia, Tolu Oni, Felix Assah, Maylene Shung-King, Safura Abdool Karim

**Affiliations:** 1SAMRC/Wits Centre for Health Economics and Decision Science—PRICELESS SA, School of Public Health, Faculty of Health Sciences, University of Witwatersrand, Johannesburg 2193, South Africa; 2Research Centre for Health through Physical Activity, Lifestyle and Sport (HPALS), Division of Physiological Sciences, Department of Human Biology, Faculty of Health Sciences, University of Cape Town, Cape Town 7935, South Africa; 3Division of Public Health Medicine, School of Public Health and Family Medicine, Faculty of Health Sciences, University of Cape Town, Cape Town 7925, South Africa; 4Health of Population in Transition Research Group, University of Yaoundé, Yaoundé P.O. Box 812, Cameroon; 5Faculty of Medicine and Pharmaceutical Sciences, University of Dschang, Dschang P.O. Box 67, Cameroon; 6MRC Epidemiology Unit, University of Cambridge, Cambridge CB2 0SL, UK; 7Health Policy and Systems Division, School of Public Health and Family Medicine, Faculty of Health Sciences, University of Cape Town, Cape Town 7925, South Africa

**Keywords:** children, adolescents, school, nutrition, physical activity, non-communicable diseases, behavior change, stakeholder engagement, priority setting, sub-Saharan Africa

## Abstract

Rates of obesity and related non-communicable diseases are on the rise in sub-Saharan Africa, associated with sub-optimal diet and physical inactivity. Implementing evidence-based interventions targeting determinants of unhealthy eating and physical inactivity in children and adolescents’ immediate environments is critical to the fight against obesity and related non-communicable diseases. Setting priorities requires a wide range of stakeholders, methods, and context-specific data. This paper reports on a novel participatory study design to identify and address contextual drivers of unhealthy eating and physical inactivity of children and adolescents in school and in their home neighborhood food and built environments. We developed a three-phase mixed-method study in Cameroon (Yaoundé) and South Africa (Johannesburg and Cape Town) from 2020–2021. Phase one focused on identifying contextual drivers of unhealthy eating and physical inactivity in children and adolescents in each setting using secondary analysis of qualitative data. Phase two matched identified drivers to evidence-based interventions. In phase three, we worked with stakeholders using the Delphi technique to prioritize interventions based on perceived importance and feasibility. This study design provides a rigorous method to identify and prioritize interventions that are tailored to local contexts, incorporating expertise of diverse local stakeholders.

## 1. Introduction

Obesity is a complex condition that results from the interplay of individual-level and socio-environmental factors related to the food, physical, cultural, and economic environments that enable or constrain healthy human behavior [1,2,3,4]. The global trend in childhood and adolescent overweight and obesity is a serious public health challenge, making the achievement of the 2030 Sustainable Development Goals to end child malnutrition (Goal 2) and improve health and well-being (Goal 3) ever more challenging. Globally, in 2016, 340 million children and adolescents aged 5–19 years were estimated to be overweight or obese [5]. Overweight and obesity are growing exponentially in low- and middle-income countries (LMICs). On the African continent, the number of overweight children aged 5–19 years increased by nearly 115% between 2000 and 2016 [5], particularly in urban settings. The term obesogenic has been increasingly used to describe the food and physical activity-built environments in LMICs that have undergone rapid urbanization processes and resulted in the high availability of cheap and convenient ultra-processed foods and reduced opportunities for physical activity through changes in infrastructure, transportation, and occupational activities [6]. Reviews of the limited literature in LMICs have established the link between obesogenic environments and behavioral and lifestyle-related risk factors of childhood and adolescent obesity and overweight, including unhealthy eating behaviors and physical inactivity [7,8,9,10,11]. Unhealthy eating, consisting of meal skipping, high consumption of ultra-processed foods, and low consumption of vegetables and fruits have been associated with high availability and marketing of cheap, unhealthy ultra-processed food and beverages in school, neighborhood, and home environments; social desirability of and prestige placed upon unhealthy food and beverage brands; peer influence; and limited time, skills, and finances of caregivers at home to prepare fresh foods [7,8,9,10,11]. Common drivers of physical inactivity included increased indoor leisure activities and entertainment, such as watching TV, playing video games, using computers and cell phones for social interaction; insecure neighborhoods; lack of open spaces in communities; and competing priorities such as schoolwork [8,9].

Despite some evidence on food and physical activity-built environments in LMICs, reviews highlighted a scarcity of data from sub-Saharan Africa [12] and the disproportionate focus on individual and interpersonal level risk factors [7] as opposed to more granular understanding of the socio-environmental drivers that shape eating and physical activity behaviors [10,11]. While reviews acknowledged global strategies aimed at developing health-promoting food and built environments [13,14], they also emphasized the need for context-specific strategies [7,10,15]. A failure to consider contextual differences in values, beliefs, norms, and environmental influences, including local and informal food vendors on eating behavior and physical activity of children and adolescents was perceived as a barrier to successful and effective implementation of global policies in LMICs (often designed for high-income settings) [7,10,11,16]. As such, reviews called for greater efforts in local evidence generation for exploring how policy could intervene more effectively to address unhealthy eating and physical inactivity environments (and consequentially behaviors) in LMICs. Reviews of existing obesity-prevention strategies targeted at children and adolescents in LMICs also highlighted certain methodological pitfalls including a lack of theoretical underpinnings of intervention design [17] and the involvement of stakeholders [9]. Studies also point out methodological approaches that have been identified as valuable, yet currently underutilized ways to establish key risk factors and prioritize impactful interventions for the prevention and control of overweight and obesity. Input from stakeholders beyond academia can enhance the relevance and quality of the research and can accelerate the uptake of research findings by decision makers and implementers [10].

To fill these knowledge gaps and to enhance the potential adoption of interventions, we developed a novel mixed-method intervention design study anchored in methods of behavior change and stakeholder engagement in two sub-Saharan African countries, South Africa and Cameroon, where child and adolescent obesity is on the rise. This paper sought to operationalize a well-known theoretical framework for participatory priority setting.

This paper describes a participatory mixed-method systematic design process to identify and prioritize interventions to improve food and built environments to support healthy eating and physical activity among children and adolescents in three sub-Saharan African urban settings: Cape Town, Johannesburg, and Yaoundé.

## 2. Study Aim and Objectives

The aim of this multi-site study was to utilize a novel method to develop intervention options and inform programmatic priorities in the immediate food and built environments of children and adolescents in and around schools and home neighborhoods (immediate environments) to support healthy eating and physical activity.

The objectives of the study were to:Identify contextual drivers of unhealthy eating and physical inactivity in children and adolescents’ immediate environments;Map evidence-based interventions to address contextual drivers of unhealthy eating and physical inactivity in children and adolescents;Prioritize evidence-based interventions to address a spectrum of underlying factors influencing unhealthy eating and physical inactivity in children and adolescents through stakeholder engagement.

To achieve these objectives, we operationalized the Behavior Change Wheel framework (BCW), a theoretical framework to understand behavior change [18] and incorporated stakeholder participation [19] to engage individuals, organizations, and communities who were directly involved in these contexts and environments and who had a direct interest in the process and outcomes of the research to create a priority setting process.

Study Rationale and Underpinning Frameworks

Several strategies exist in LMICs to support healthy dietary and physical activity behaviors of school-aged children and adolescents; however, evidence of their effectiveness in improving anthropometric (body mass index and weight status) outcomes is limited [12,16]. Changing behavior is a challenging task; effective interventions need to be underpinned by theory- and evidence-based frameworks that incorporate an understanding of the behavior to be changed. The BCW provides a theoretical basis and mechanism to identify factors influencing behavior and interventions that may be used to address these factors [18]. In this study, we drew on the BCW framework (Figure 1) developed by Michie et al. (2014) and incorporated stakeholder engagement activities as key components of the design to inform priority setting. The BCW framework is grounded in behavior change theoriesas it was developed by a cross-disciplinary team of researchers after analyzing 83 related theories or frameworks and over 1500 constructs. It is linked to evidence-based intervention functions that can orient an intervention to a targeted setting and population of interest. The BCW has found widespread application in several health policy domains, including in diet and physical activity policy [20]. The BCW, centered around the COM-B model, envisages that for any behavior to happen, the individual and/or population must have the necessary capability, opportunity, and motivation (Figure 1 inner circle) to undertake the desired behavior.

Surrounding the inner core is a second layer (second circle on Figure 1), comprising nine intervention functions. These showcase various ways through which a behavior can be changed (Education, Persuasion, Incentivization, Coercion, Training, Enablement, Modelling, Environmental Restructuring and Restrictions). Lastly, the outermost layer of the BCW identifies seven policy levers (Environmental/Social planning, Communication/Marketing, Legislation, Service provision, Regulation, Fiscal measures, and Guidelines). These levers can be used to target different intervention functions. Given the close links between policy and behavior change outlined in the framework as well as the holistic view the framework provides for behavior change, the BCW offers a particularly appropriate lens to map existing interventions and identify new interventions. Collectively, the interaction of these various layers is likely to elicit a behavioral shift. Critically, the BCW model recognizes the role that systems and communities play in individual choices and postulates that it is necessary to adopt interventions to address all three components (capability, opportunity, motivation) to change behavior. In this study, the adaptation of this model occurred through stakeholder engagement, a method that has been recognized as currently neglected but critical for the success of behavior change research [21] and that has been identified as a specific weakness in the implementation of the BCW approach [22]. To fill this scientific gap, efforts were made to incorporate activities to formally integrate the perspectives of individuals, organizations, and communities throughout the research process. 

## 3. Materials and Methods

The study was a collaboration between researchers from South Africa (Johannesburg and Cape Town), Cameroon, and the United Kingdom (UK) (Cambridge) as part of the Global Diet and Activity Research Network (GDAR Network) that aims to contribute to the field of diet and activity research in African and Caribbean countries and to inform policies for NCD prevention [23]. A virtual protocol development workshop with technical oversight provided by the University of Cambridge was conducted in June 2020 with the study teams across three sites. This protocol was adapted for each site as needed, while retaining the main research questions and methodological approaches.

### 3.1. Study Settings and Design

The study was conducted in three settings, two located in South Africa (Johannesburg and Cape Town) and the third in Cameroon (Yaoundé). This study utilized mixed methods comprising three distinct phases, underpinned by an iterative scientific process (Figure 2): (1) secondary analysis of existing qualitative data [24,25] on children and adolescents’ immediate food and physical activity environments collected as part of the GDAR Network research (Table 1); (2) intervention identification and mapping through a review of the World Cancer Research Fund International’s NOURISHING [26] and MOVING [27] databases; and (3) prioritization of interventions with relevant stakeholders using Delphi methods [28].

### 3.2. Phase One—Identify Drivers in Behavioral Terms

In the first phase, a secondary analysis of qualitative data [29] (collected as part of a GDAR Network research study exploring multi-level levers to support healthy eating and active living [24,25]) was carried out to explore contextual drivers of unhealthy eating and physical inactivity of primary school-aged children (6–11 years) in Johannesburg, and adolescents (10–19 years) in Cape Town and Yaoundé (Table 1).

Data analyzed included interviews at the different sites with relevant stakeholders as outlined in Table 1. During the interviews (that were recorded and transcribed verbatim), participants were asked to reflect on aspects of the food and physical activity environments in their school and home neighborhood and provide suggestions for improvements. In addition to the interviews, researchers in Cape Town and Yaoundé facilitated thematic analysis workshops with adolescents from high schools to identify barriers and facilitators of healthy eating and physical activity. During these workshops, students were asked to suggest and prioritize solutions to overcome the barriers.

The data sources were analyzed to investigate questions beyond the objectives of the primary study. As a result, there is a level of heterogeneity in the data sources relied upon [24,25]. However, the method and data analysis techniques were uniform across the study sites. The analysis involved reading the transcripts and systematically noting drivers of unhealthy diets and physical inactivity considering the behavioral, social, and environmental contexts in which the behaviors occur and the individual, group, or population levels of focus. 

Each interview transcript was inductively coded, and thereafter codes with similar content were grouped together under specific themes. At each site, coding was performed independently by two researchers with constant comparison of codes. Any discrepancies in the coding process were resolved by discussion during monthly meetings between coders and other researchers in the study teams. Each study team used qualitative analysis software such as MAXQDA 2020 (VERBI Software, 2019) [30] and NVivo version 12.0 [31] to facilitate analysis. Once data were coded, researchers across all sites developed and agreed upon a common data extraction matrix in Microsoft Excel (Appendix A).

When populating the matrix, researchers extracted data using the BCW domains on barriers and facilitators, focusing on children and adolescents’ capability, opportunity, and motivation to obtain a healthier diet and engage in physical activity. The common matrix facilitated summarizing the findings at each site. Data extraction training was conducted with technical support from study team members with previous experience in using the BCW framework. This ensured consistency across the sites and necessary adjustments were made. The data extraction matrix was extensively pilot-tested, checking cross-setting consistency on understanding meaning and quality control achieved by having one or more common topic experts at data extraction training and testing sessions.

### 3.3. Phase Two—Intervention Mapping

In phase two, the contextual drivers of unhealthy eating and physical inactivity identified in phase one were used to explore evidence-based interventions that might help to improve the school and home neighborhood food and physical activity environments. To map existing evidence-based interventions in the literature, we relied on the NOURISHING database for diet and nutrition policy actions, and the MOVING database that complements the NOURISHING database with respect to physical activity policy actions. The NOURISHING and MOVING databases identify policy areas where governments can intervene to promote healthy diets and physical activity. Both databases collect policy actions from around the world implemented at a national level and which were deemed effective at the time of the study. Research team members reviewed the NOURISHING and MOVING interventions and identified those most relevant to address site-specific phase one drivers of unhealthy eating and physical inactivity in schools and home neighborhoods. Interventions that were perceived as too broad to address a specific driver were modified by each site, guided by the proposed solutions from the stakeholders in phase one that could directly address the driver within its specific context (see Table 2 for examples). 

We associated each intervention with the most dominant intervention functions (see second circle in Figure 1) and a policy category (see outermost circle in Figure 1). To do this, we relied on the BCW’s established linkages between intervention functions and associated policy categories. For example, if the intervention function of restriction is selected to change a target population’s behavior (i.e., prioritization of physical activity and healthy eating in schools) policy categories such as regulations or guidelines can potentially deliver that intervention. When mapping the interventions, care was taken to ensure that: (a) there was a diversity of intervention types to address multiple drivers; and that (b) these interventions were responsive to the components of the BCW driving the behavior targeted by the intervention. From this procedure, each site identified possible interventions for each of the BCW components to be prioritized by stakeholders in phase three of the study. Because behavior change processes exist in challenging complex food and built environment systems, single and simple interventions are unlikely to result in long-term changes in behavior. Hence, each site identified interventions operating on multiple levels of influence within the COM-B structure and targeting different factors influencing behavior.

### 3.4. Phase Three—Prioritizing Interventions

In the third phase, we applied a variation of the Delphi method [28] to solicit stakeholder opinions on priority interventions with a potential to improve the food and built environments in schools and home neighborhoods over several iterative rounds of surveys. The Delphi approach is a widely used and recommended method to obtain consensus among participants with domains of expertise. While researchers have developed variations of the Delphi method since its development in the 1950s, it is commonly known as an iterative process that uses a systematic progression of repeated rounds of questionnaires to collect data from a panel of selected individuals and determine group consensus [32]. This study falls within the “ranking-type” variant with two or more survey rounds using two concurrent rating scales, including *importance* and *feasibility*, and survey rounds using a scale of *priority*.

There is no agreement regarding the size of the panel in a Delphi exercise. The literature indicates that panel sizes vary from a few to hundreds and highlights that meaningful results can be obtained with as few as 10–15 participants [33,34]. In this study, each site purposely identified stakeholders whose work and interests were likely to be relevant in identifying strategies that address the contextual drivers of unhealthy eating and physical inactivity in children and adolescents. The point of departure was identifying stakeholders engaged in the primary studies [24,25] utilized in phase one of the study. This was complemented with a targeted desk-based review to extend the list of potential Delphi participants. Stakeholder panels consisted of key decision makers (i.e., educational authorities and school principals, school governing bodies) and other stakeholders (i.e., parents, adolescents, school vendors, community organizations, etc.). 

Once panel members were contacted, the goals and processes of the project were explained via a virtual meeting for South African participants and through face-to-face meetings in Cameroon; consent and assent were obtained to participate in the Delphi exercise. Thereafter, participants were forwarded a link to an online survey in Google Forms to complete several rounds of surveys in South Africa. In some cases, where participants were not contactable via email, a member of the research team approached them in person, and a hard copy of the questionnaire was self-administered, or administered telephonically, then captured electronically. In Cameroon, all surveys were administered during a face-to-face workshop. The differences in site-specific implementation of the Delphi exercise, including the channels and platforms used for interaction (virtual vs. in-person) and the number of interactions with stakeholders were due to site-specific containment measures during the COVID-19 pandemic, as well as different cultural norms and expectations from stakeholder engagement.

The Delphi exercise consisted of quantitative survey rounds in each site. Delphi participants were given one to four weeks to respond to each round of the survey. In the survey, we asked participants to separately rank phase two interventions according to their perceived feasibility and importance using a five-point Likert Scale: “Not at all important” (1), “Low importance” (2), “Neutral” (3), “Somewhat important” (4), and “Very important” (5). The same scoring applied for perceived feasibility of the interventions.

Between the Delphi rounds, responses from earlier rounds were analyzed and used to narrow the initial list of interventions for subsequent rounds of the Delphi. The group response median value and the quartile deviation (QD) were used as a reference for the degree of importance, feasibility, and consensus. This is consistent with previous Delphi studies [23,25,26]. The collective median and QD of each rating were calculated for the whole group in Johannesburg and Cape Town. In Yaoundé, a sub-group analysis was conducted where responses were analyzed by stakeholder group category, and the top-rated interventions for importance and feasibility were selected to constitute a narrowed list for subsequent rounds, specific to each stakeholder group.

In the final round, participants were asked to review the narrower list of interventions resulting from earlier rounds, rank order them to establish priorities among stakeholders and determine consensus for the final ranked list of interventions. 

To determine the overall ranking of final round interventions, we used either stakeholder group consensus (in the face-to-face workshop in Cameroon) or the Borda count method [35] (in South African sites), an approach to aggregating individual ranked preferences. The ranking was 1 to *n* (number of interventions in the final round). We counted how many times each intervention was ranked 1 to *n* by all participants. We multiplied this number by the ranking number (1 to *n*) and then added this up to determine the total Borda count. The interventions with the highest Borda counts were the ones that were considered as ranking highest.

## 4. Reflections on the Study Design and Methods

The primary objective of the multi-site study was to explore factors driving behaviors in particular contexts and utilize theoretical and participatory research methods in a novel way to identify and prioritize context-specific interventions to address these factors and to improve the food and built environments in the three sites. The strength of this study design, in particular, lies not only in the identification of context-specific drivers of unhealthy behaviors but also in the engagement of stakeholders in the research and the use of a common design process in multiple settings. This ensures that the site-specific interventions that emerged are tailored to the unique complexities of each setting and will be invaluable in guiding policies to improve demand for and access to healthy food and opportunities for physical activity. 

Our study design also fills a gap in the food and built environment literature. Research on this topic in LMICs is limited, particularly from sub-Saharan Africa [11]. A recent systematic review of food environment studies revealed that from 70 articles, only 11 were from sub-Saharan Africa with 7 of these from South Africa, and only 4 studies documented school food environments. Quantitative studies dominated the research literature with little focus on the qualitative process of evidence generation. A mixed-methods approach, such as the present study, has rarely been utilized despite its potential to better capture real-life contextual understandings and multilevel perspectives through the triangulation of various types of evidence [3].

Another strength of our study lies in the extensive engagement of stakeholders at multiple points throughout the project, beginning with the qualitative interviews for secondary data analysis which sought to understand underlying drivers and then concluding with these and other stakeholders prioritizing interventions for their settings. Stakeholder engagement in priority setting enabled the necessary tailoring of policy actions to local contexts, increasing their relevance and feasibility. Being able to engage the same group of stakeholders across the lifetime of the project might result in increased buy-in and enhance the likelihood of uptake of research findings [36]. Moreover, building consensus around priority interventions using the Delphi approach provided research teams with a sense of confidence that the recommendations from the study had a high level of endorsement from the stakeholders who would either implement or be subject to the interventions themselves.

Project-related logistical constraints strongly influenced the overall study design, including stakeholder engagement strategies. A key constraint to the development and implementation of the stakeholder engagement included the relatively short time frame of the project (12–17 months) and project personnel challenges (lack of time, turnover). The short time frame was further constrained by the timing of engagement with certain stakeholder groups, such as school staff whose availability was determined by school holidays. For example, school staff were not available during breaks and around the time when schools reopened. The coinciding COVID-19 pandemic within the study timeframe also meant that policy makers, especially from the Department of Health were less available and willing to dedicate their time to engage with the research teams. We overcame some of these challenges by continuous follow up through emails and phone calls to participants and in-person school visits.

While a shared protocol across sites was viewed as a best-case model, site-specific execution of the proposed methodology necessarily varied due to local contextual factors. This further underscores the context-specific nature of the study. First, reliance on site-specific secondary data sources resulted in differences in the scope of determinants and types of stakeholders engaged at each site. While Cape Town and Yaoundé had information on physical activity and the built environment, Johannesburg’s data was limited to the food and beverage environments. Cape Town and Yaoundé engaged with adolescents and caregivers, while the primary study that informed phase one in Johannesburg documented the perspectives of school staff. 

## 5. Conclusions

Achieving Goals 2 (to end child malnutrition) and 3 (improve health and well-being) of the 2030 Sustainable Development Goals will require improvements in diet and physical activity of children and adolescents. Food and built environments in schools and home neighborhoods are significant to achieving these goals in sub-Saharan Africa. This paper describes the development of a novel method to adapt and operationalize an existing theoretical model to identify and prioritize interventions that are tailored to context-specific challenges and represent the perspectives of local stakeholders.

## Figures and Tables

**Figure 1 ijerph-19-10263-f001:**
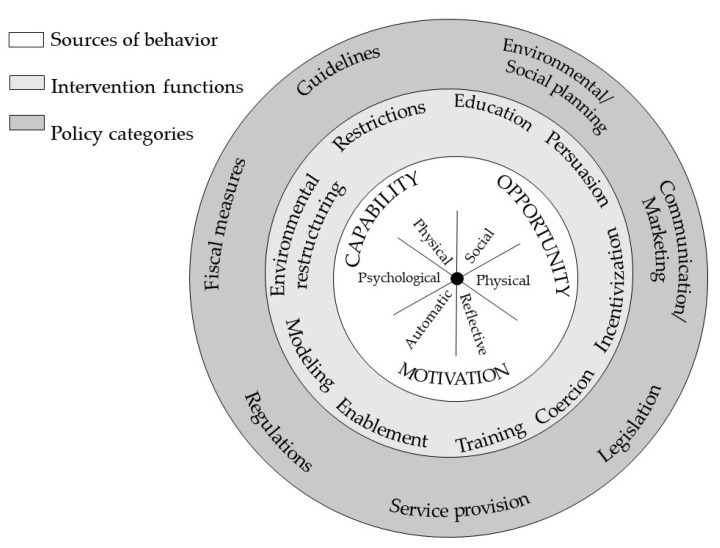
Behavior Change Wheel. Reproduced with permission from Michie S, Atkins L, West R. (2014) *The Behavior Change Wheel: A Guide to Designing Interventions*. London: Silverback Publishing. www.behaviourchangewheel.com, (accessed on 1 May 2022).

**Figure 2 ijerph-19-10263-f002:**
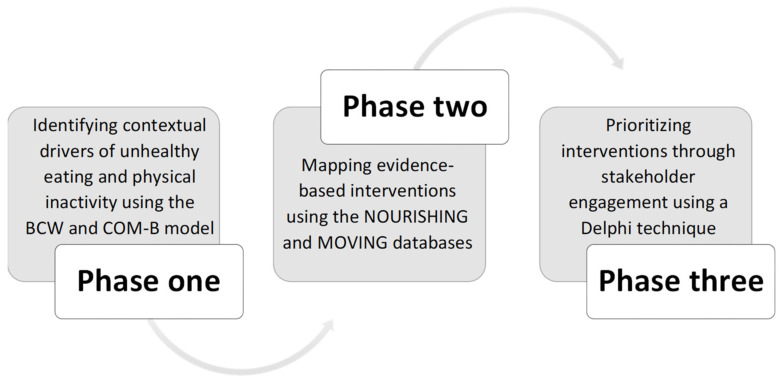
Flowchart outlining the three-phase iterative study design.

**Table 1 ijerph-19-10263-t001:** Overview of site-specific data used in secondary analysis of qualitative data.

Study Site	Stakeholders	Data Collection Tool	Description of Data
Johannesburg, South Africa	Public primary school principals, school tuck shop * owners and managers, heads of school governing bodies	Face-to-face qualitative interviews using a semi-structured interview guide	Issues covered: policies and guidelines related to the food environment, perceptions of the food and beverage environment, nutrition-related support/activities available at school, facilitators and barriers to change the food and beverage environment.
Cape Town, South Africa and Yaoundé, Cameroon	High-school-going adolescents from low and middle/high income communities and their parents	Qualitative telephonic in-depth interviews	Issues covered: food procurement, storage and preparation, food choices and challenges, leisure time activities, and the meaning and significance of food and physical activity within families and of adolescents.
	High-school-going adolescents	Mobile application	Photos, texts, and audio narratives of adolescents’ neighborhood, school, home environments and journey from home to school with a focus on food outlets, food and sugar-sweetened beverage adverts, physical activity opportunities that are perceived to have an impact on adolescents’ diet and physical activity.

* Designated areas within the school premises that have food and beverage items.

**Table 2 ijerph-19-10263-t002:** Examples of interventions resulting from the intervention mapping.

Driver	COM-B Domain	NOURISHING-MOVING		BCW Policy Category	BCW Intervention Function	Context Specific Intervention
		Policy Area	Intervention			
Rewarding children with unhealthy foods	Opportunity (Social) Motivation (Automatic)	Offer healthy food and set standards in public institutions and other specific settings	Mandatory standards for food available in schools including restrictions on unhealthy food	Regulation	Restriction	Stop the use of food as a reward in schools
Cheap and unhealthy foods and beverages at school tuck shop; influence of product characteristics, i.e., shelf life and pricing	Opportunity (Physical) Motivation (Reflective)	Use economic tools to address food affordability and purchase incentives	Targeted subsidies for healthy food	Fiscal measures	Incentivization	Incentivize school tuck shops to sell healthy food and drinks by giving subsidies or decreasing tax
Brand recognition of unhealthy foods	Opportunity (Social) Motivation (Reflective)	Restrict food advertising and other forms of commercial promotion	Mandatory regulation of food marketing in schools and more broadly	Legislation	Persuasion; Environmental restructuring	Stop advertising of unhealthy food products to children, including promotional materials, billboards, or signs in the school and surrounding areas
Polluted neighborhoods, unclean and vandalized public equipment, and PA facilities	Opportunity (Physical)	Visualize and enact structures and surroundings which promote physical activity	Policies that support access to quality public open space and green spaces	Environmental/social planning	Environmental restructuring, Enablement	Encourage strong community participation and engagement with local government to prevent vandalism, reduce litter, and promote upkeep of public spaces
Limited sports equipment in the schools and few sports options available to participate	Opportunity (Physical) Motivation (Reflective)	Make opportunities and initiatives that promote physical activity in schools, the community, and sport and recreation	Financial and non-financial incentives to promote physical activity	Service provision	Enablement	Collaboration between schools and sports clubs for student access free of charge and donations of sports equipment
Physical education and activities not taken seriously by teachers and learners	Motivation (Reflective)	Normalize and increase physical activity through public communication that motivates and builds behavior change skills	Develop and communicate physical activity guidelines	Guidelines	Education, Incentivization	Co-creation of creative ways of PE varieties by both the teachers and the learners based on the PE curriculum to motivate learners to be more physical active
Lack of motivation and laziness to exercise at home or in the neighborhood	Opportunity (Social) Motivation (Automatic)	Normalize and increase physical activity through public communication that motivates and builds behavior change skills	Mass communication campaigns including social marketing to increase awareness and knowledge about benefits of physical activity through the life course	Communication/marketing	Modelling, Persuasion	Social media campaigns and mass media messaging to raise awareness of the health benefits of playing sports or regular physical activity through celebrity endorsement; role models in various fields, such as athletes, singers, and actors.

## Data Availability

Not applicable.

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
