# Peer review of "A Mixed-Methods Participatory Intervention Design Process to Develop Intervention Options in Immediate Food and Built Environments to Support Healthy Eating and Active Living among Children and Adolescents in Cameroon and South Africa"

_ijerph, 2022, doi:10.3390/ijerph191610263_

Round 1

Reviewer 1 Report

Overall, this paper is a valuable contribution to the literature about the process of mixed-methods research supporting improvements to healthy food and physical activity environments. The study includes unique research methods, including behavior change theory and community engagement processes. In addition, the research was conducted in an understudied geographical location.

The highest priority area for improvement is the conclusion. The conclusion includes Sustainable Development Goals 2030 that were not previously mentioned in the research as priorities. The conclusion should be re-written so that it reflects the goals as stated in the abstract and introduction of the paper.

A top priority would be in the addition of more results to include in section 4, line 297. Although this paper is focused on the process, this reviewer would like to have additional concrete examples of how the “final product”/site-specific interventions based on BCW adaptations were created. This could be presented in a table. In addition, another table to present summaries of the site-specific processes might help elucidate the methods more clearly. E.g., the reviewer is interested in how many experts were included in the Delphi method per site.

Another priority area would be in explaining the differences in methods for each research site earlier on in the manuscript. The differences due to the pandemic (E.g. Zoom vs in-person interviews mentioned in lines 349-352) should have been explained earlier. This would have helped the reader understand why there were inconsistent methods employees. To this reviewer, difference in methods were thought to be due to cultural differences/level of familiarity with technology/access to technology in different countries.

Specific suggestions:

Line 176, Table 1: Suggest to add the location of study site for every section. It is unclear if the “High-school going adolescents from low and middle/high income….” is in Johannesburg or not.

Line 176, Table 1: Please explain what a tuck shop in the first instance it is mentioned in the paper.

Lines 189-197: In this paragraph, did the investigators have two researchers independently extract data from the same interview, then compare results to measure consistency? If yes, it is unclear that this quality assurance method was employed.

Line 218- Table 2. Define what “taps” are. This reviewer is guessing that it is a water fountain for drinking water, but is unsure.

Line 220-221: This reviewer thinks that this sentence is a run-on. Please re-word or split the sentence in two.

Line 452, reference 28 seems to need the last names of article authors spelled out.

Reviewer 2 Report

The article is well-written and describes well the design process of a project. However, the article lacks criticality and has not provided enough evidence to establish the contribution of the article. Simply describing the process of a research project may not be enough for publication and the authors should address the concerns in the attached file. The main comment is that the introduction section should review existing evidence on the topic and highlight the need for a novel methodological approach which the authors claim to have generated here. I also wonder whether the authors can  present  graphically how they adapted the COM-B model.
